# Running wheel exercise induces therapeutic and preventive effects on inflammatory stimulus-induced persistent hyperalgesia in mice

Cesar Renato Sartori[ID]*, Marco Pagliusi, Jr[ID], Ivan José Magayewski Bonet[ID], Claudia Herrera Tambeli, Carlos Amilcar Parada

Department of Structural and Functional Biology, State University of Campinas, Campinas, SP, Brazil

* sartoric@unicamp.br

**Data Availability Statement:** All data are within the paper and its Supporting Information files.

## Abstract

Chronic pain affects significant portion of the world's population and physical exercise has been extensively indicated as non-pharmacological clinical intervention to relieve symptoms in chronic pain conditions. In general, studies on pain chronification and physical exercise intervention have focused on neuropathic pain, although chronic pain commonly results from an original inflammatory episode. Based on this, the objective of the present study was to investigate the therapeutic and preventive effect of the running wheel exercise on the persistent hyperalgesia induced by repetitive inflammatory stimulus, a rodent model that simulates clinical conditions of chronic pain that persist even with no more inflammatory stimulus present. To evaluate the therapeutic effect of physical exercise, we first induced persistent hyperalgesia through 14 days of PGE2 hind paw injections and, after that, mice have access to the regular voluntary running wheel. To evaluate the preventive effect of physical exercise, we first left the mice with access to the regular voluntary running wheel and, after that, we performed 14 days of PGE2 hind paw injection. Our results showed that voluntary running wheel exercise reduced persistent mechanical and chemical hyperalgesia intensity induced by repetitive inflammatory stimulus. In addition, we showed that this therapeutic effect is long-lasting and is observed even if started belatedly, i.e. two weeks after the development of hyperalgesia. Also, our results showed that voluntary running wheel exercise absolutely prevented persistent mechanical and chemical hyperalgesia induction. We can conclude that physical exercise has therapeutic and preventive effect on inflammatory stimulus-induced persistent hyperalgesia. Our data from animal experiments bypass placebo effects bias of the human studies and reinforce physical exercise clinical recommendations to treat and prevent chronic pain.

## Introduction

Chronic pain affects significant portion of the world's population. In the USA, for example, one third of the population suffers from chronic pain [1]. In addition, according to the World

**Funding:** CRS, CAP and CHT was supported by the São Paulo Research Foundation (FAPESP - http://www.fapesp.br) #2011/23531-6 and #2012/08922-1. MPJr, and IJMB was supported by Coordenação de Aperfeiçoamento de Pessoal de Nível Superior - Brasil (CAPES - https://www.capes.gov.br) - Finance Code 001. The funders had no role in study design, data collection and analysis, decision to publish, or preparation of the manuscript.

**Competing interests:** The authors have declared that no competing interests exist.

Health Organization, musculoskeletal disorders, typically characterized by chronic pain, is the most common cause of disability in the world. This highlights the great economic and social impact of chronic pain in the world, generating intense suffering for the individual and a great burden on public health.

It is important to establish that pain perception is part of a repertoire used by animals (including humans) to help them cope with life-long adversity [2]. Thus, acute pain promotes clear protection to the survival of the organism [2]. On the other hand, in chronic pain the adaptive integration between sensory and emotional processes is extremely disturbed, that is, pain persists for a long time even with no threat to the individual in a maladaptive phenomenon [3]. Considering the complexity of chronic pain pathology, we can expect that its treatment must also be complex. In fact, treatments for chronic pain have multifactorial components, although most patients are not referred for multidisciplinary treatment in the early onset [4]. Moreover, pharmacological approach to treat chronic pain conditions have limited effectiveness, often showing undesirable side effects, in addition to abuse and addiction risk and even related deaths [4, 5]. Therefore, multidisciplinary programs to treat chronic pain, including non-pharmacological approaches, are necessary.

One of the most commonly suggested non-pharmacological therapies in chronic pain treatment is physical exercise (for review see [6 and 7]). In fact, physical exercise has been extensively indicated as non-pharmacological clinical intervention to relieve symptoms in chronic pain conditions such as low back pain, rheumatic diseases, migraine, and fibromyalgia [8–14]. In animal models of chronic pain, physical exercise has also shown therapeutic effect, although most of the studies were performed in neuropathic pain models such as nerve injury [15–21].

In general, studies on pain chronification have focused on neuropathic pain, although chronic pain commonly results from an original inflammatory episode [22]. Based on this, Ferreira *et al.* [23] developed an animal model which simulates chronic pain triggered from an inflammatory event. In this model, intraplantar injections of prostaglandin E2 ($PGE_2$) is daily administred in rodents for 14 consecutive days (repetitive inflammatory stimulus) and, after the end of the injections, rodents show persistent hyperalgesia up to 30 days. This model simulates clinical conditions of chronic pain that persist even with no inflammatory stimulus present [23, 24].

Based on the abovementioned, the objective of the present study was to investigate the therapeutic and preventive effect of the running wheel exercise on the persistent hyperalgesia induced by repetitive inflammatory stimulus ($PGE_2$) in mice. A therapeutic and preventive approach is relevant because most of the studies aim to investigate only the therapeutic effect of the physical exercise. Moreover, the animal model used in the present study also allows us to evaluate the preventive effect of physical exercise on pain chronification.

## Methods

### Animals

We used eight weeks old C57BL/6J male mice (25g) obtained from the Multidisciplinary Center for Biological Investigation (CEMIB) at the State University of Campinas (UNICAMP). Mice were housed under standard laboratory conditions (22±1˚C, 12-hour light/dark cycles, food and water *ad libitum*). Animal care and handling procedures were in accordance with International Association for the Study of Pain (IASP) guidelines for the use of animals in pain research and the protocols for animal care and use were approved by the institutional Committee for Ethics in Animal Experimentation at UNICAMP (CEUA/IB-UNICAMP), São Paulo, Brazil, protocol number 2857–1. All effort was made to limit the number of animals used.

## Experimental design

In the present study, experiments were performed to investigate the physical exercise effect on persistent hyperalgesia induced by prostaglandin $E_2$ ($PGE_2$). Experiment 1 and 2 was performed to investigate the therapeutic effects of exercise on persistent hyperalgesia, while experiment 3 was performed to investigate the preventive effect of physical exercise.

**Experiment 1.**   In the Experiment 1, mice were firstly individualized, remaining single housed for the entire experimental protocol. Following, mice submitted to persistent hyperalgesia induction through $PGE_2$ intraplantar injections (or saline in control condition) in the hindpaw for 14 consecutive days (details described below). After that, mice had free access to a running wheel for 28 days (or remaining sedentary in control condition). Over the exercise period mice were tested for mechanical hyperalgesia by the electronic von Frey method (details described below). Finally, mice were tested for nociceptive response to chemical stimuli (details described below) at the end of the experimental period. Experiment 1 had the following experimental groups: sedentary/saline, sedentary/$PGE_2$, exercise/saline, exercise/$PGE_2$. Graphical diagram of the experimental design of experiment 1 is presented on **Fig 1A**.

**Experiment 2.**   The Experiment 2 was divided into two sets of experiments: A and B. On Experiment 2A, mice were firstly individualized, remaining single housed for the entire experimental protocol. Following, mice were submitted to persistent hyperalgesia induction through $PGE_2$ intraplantar injections (or saline in control condition) in the hindpaw for 14 consecutive days (details described below). After that, mice had free access to a running wheel for just 14 days, returning to sedentary condition for more 14 days after the running period. This experiment was performed to evaluate if even shorter time (14d) in exercise would be positive to the mice and to evaluate if the exercise effects would be long-lasting. On Experiment 2B, mice were firstly individualized, remaining single housed for the entire experimental protocol. Following, mice were submitted to persistent hyperalgesia induction through $PGE_2$ intraplantar injections (or saline in control condition) in the hindpaw for 14 consecutive days (details described below). After that, mice remained on sedentary condition for 14 days andonly after that had free access to a running wheel for only 14 days. This experiment was performed to evaluate if the exercise would be positive even introduced belatedly, after a longer time of persistent hyperalgesia. Mice were tested for mechanical hyperalgesia on several timepoints and were tested for nociceptive response to chemical stimuli at the end of the experimental period. Experiment 2 had the following experimental groups: Saline/Exercise 14d-28d, $PGE_2$/Exercise 14d-28d, Saline/Exercise 28d-42d, $PGE_2$/Exercise 28d-42d. Graphical diagram of the experimental design of experiment 2 is presented on **Figs 3A and 4A**.

**Experiment 3.**   In the Experiment 3, mice were firstly individualized, remaining single housed for the entire experimental protocol. Following, mice had free access to running wheel and were submitted to persistent hyperalgesia induction through $PGE_2$ intraplantar injections (or saline in control condition) after 7 days of exercise (starting on day 8). Mice remained with free access to the running wheel for more 21 days, also totaling 28 days of exercise. Mice were tested for mechanical hyperalgesia on several timepoints (**Fig 6A**). Finally, mice were tested for nociceptive response to chemical stimuli at the end of the experimental period. Experiment 3 had the following experimental groups: sedentary/saline, sedentary/$PGE_2$, exercise/saline, exercise/$PGE_2$. Graphical diagram of the experimental design of experiment3 is presented on **Fig 6A**.

## Voluntary running wheel exercise

Mice from exercised groups had free access to an individual running wheel 24h/day. It is worth mentioning that all mice remained single housed during all experimental protocol and

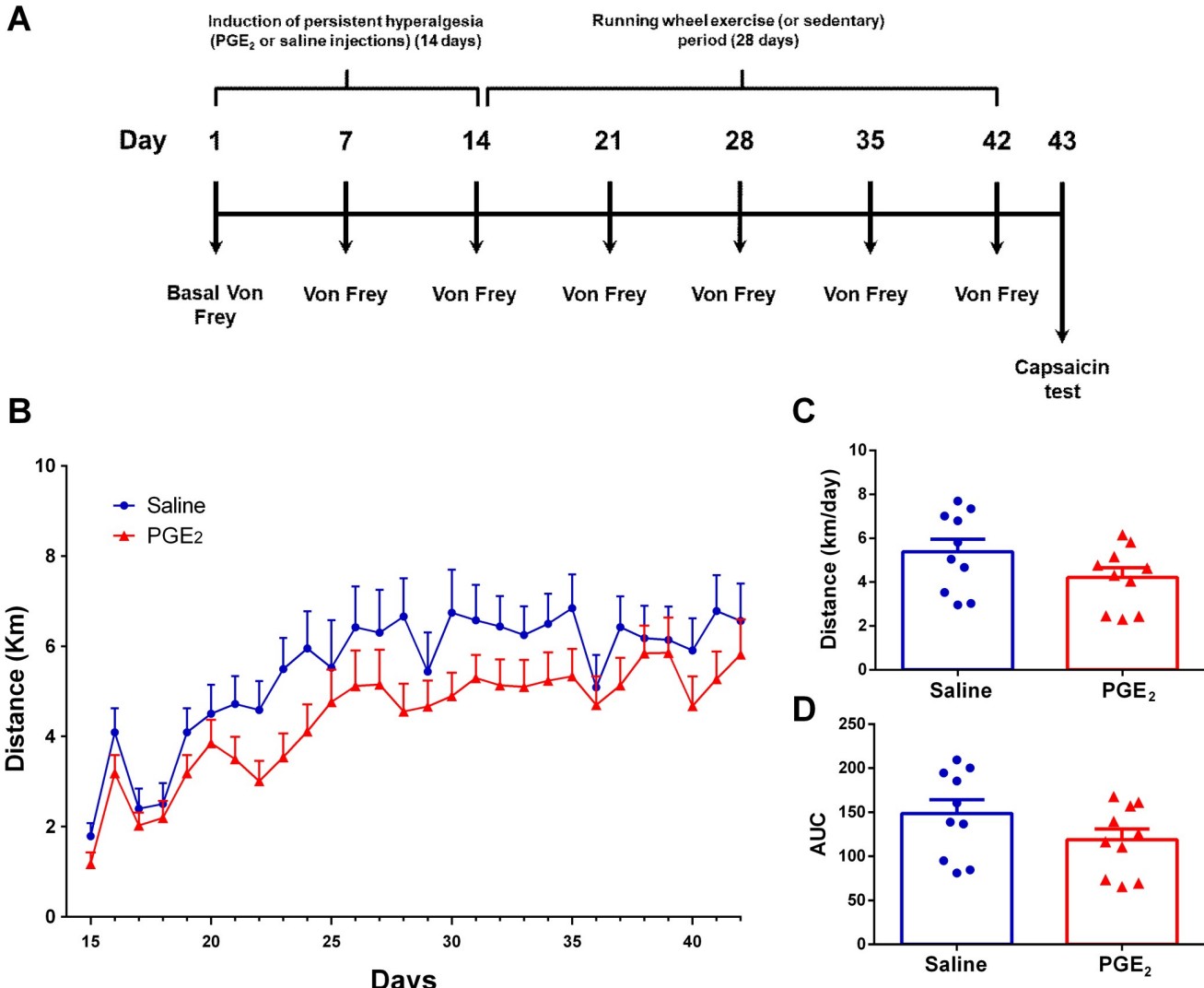

**Fig 1. Mice activity on running wheel. (A)** Graphical diagram of the experimental design **(B)** Average distance (km) traveled on each day over 28 days of running wheel free access. **(C)** Average distance (km) traveled per day on all 28 days of running wheel free access. **(D)** Area under the curve (AUC) for each group calculated from (A) data. There were no significant differences between the groups. Data are showed as mean ± SEM (N = 10 per group). Two-way ANOVA followed by Sidak posttests (B); Unpaired t test (C) and (D).

each exercised mouse had his own running wheel. The running wheel was plastic, with 9.2cm in diameter, and had a magnetic sensor that was connected to computer software to record summation of wheel revolutions within five-minute intervals (Columbus Instruments, OH, USA). The distance traveled by exercised mice on running wheel was recorded 24 h/day and was quantified and expressed in kilometers.

## Persistent hyperalgesia induction model

For persistent hyperalgesia induction, we used a mouse model of repetitive inflammatory stimulus intraplantar injection previously described by Ferreira *et al.* [23] and Villarreal *et al.* [24]. Briefly, we injected prostaglandin $E_2$ (PGE$_2$, 90ng/25μl/paw, Sigma, USA,) or vehicle (saline) subcutaneously on the plantar surface of the left hind paw, once a day over 14 consecutive

days. As originally described in this model, after 14 consecutive days of PGE$_2$ injections mice show long-lasting mechanical hiperalgesia.

## Electronic von Frey test

To evaluate the mechanical nociceptive threshold we performed the electronic von Frey test [25]. Before the beginning of the nociceptive test, mice were placed in acrylic cages measuring 12x20x17cm, with floor consisting of a 5mm$^2$ mesh of non-malleable wire with 1mm of thickness. Mice remained in these cages during 30 minutes for habituation. The electronic von Frey test consisted of evoking a hind paw flexion with a hand-held force transducer adapted with a 0.5mm polypropylene tip (electronic von Frey; Insight LTDA, SP, Brazil). A tilted mirror placed under the grid provided a clear view of the mouse hind paw. The experimenter was trained to apply a pressure on the plantar surface of the mouse hind paw with a constant gain of strength until the mouse withdraws the paw in a clear flinching response. The stimulus is automatically discontinued, and its intensity was recorded when the paw withdrawn. The results are shown as the variation of the nociceptive threshold in grams (gram-force) obtained by subtracting the average of three values observed in the basal von frey test from those of the average of three values obtained in each of all following tests performed (Δ mechanical nociceptive threshold in grams). This variable was used to represent mechanical hyperalgesia intensity. The investigator was blind to all experimental treatments.

## Capsaicin test

Capsaicin test was performed to evaluate the chemical hyperalgesia intensity. For habituation, 30 minutes before starting the experiment mice were placed in the capsaicin test apparatus, consisted in an observation box measuring 30x30x30cm with of one clear glass wall at the front, three mirrored walls, and a mirrored floor. Following the habituation period, we injected capsaicin in the subcutaneous tissue of the left hind paw (0.1μg/15μl/paw). The nociceptive response was characterized by the act of lifting the paw reflexively (flinching) and was quantified during a 5 min period. The investigator was blind to all experimental treatments.

## Statistical analysis

Statistical analysis of the data obtained was performed in GRAPH-PAD PRISM version 6.00 for Windows (GraphPad Software, San Diego, CA, USA). The results are shown as the mean ± standard error of mean (SEM). Statistically significant difference was considered when p ≤ 0.05. As specified in the figures legends, the following analyzes were used: Two-way ANOVA followed by Sidak multiple comparison posttests, One-way ANOVA followed by a Bonferroni multiple comparison post test, and Unpaired T tests.

# Results

## Experiment 1

**Repetitive prostaglandin E2 injections did not affect mice activity in the running wheel.** **Fig 1** shows the Experiment 1 experimental design and the mice activity on running wheel. Two-way ANOVA statistical analysis demonstrated significant time effect ($F_{1,18}$ = 18.48; P ≤ 0.0001) but not group effect ($F_{1,18}$ = 2.253; P ≤ 0.1507) considering the average distance run on each day over the 28 days of running wheel free access (**Fig 1B**). The *post hoc* Sidak multiple comparison test revealed no significant differences between groups considering the average distance run. Also, considering the average distance run per day on the total period (**Fig 1B**), statistical analysis showed no significant differences between groups (Saline:

5.390 ± 0.5725 km/day; $PGE_2$: 4.215 ± 0.4422km/day; P = 0.1217; t = 1.624, df = 18). **Fig 1C** shows the area under the curve (AUC) and statistical analysis also showed no significant differences between groups (Saline: 148.7 ± 15.54; $PGE_2$: 118.8 ± 12.25; P = 0.1479; t = 1.512, df = 18).

**Running wheel exercise reduced persistent hyperalgesia intensity.** **Fig 2A** shows the variation of the mechanical nociceptive threshold in grams (Δ mechanical nociceptive threshold), representing the hyperalgesia intensity. Two-way ANOVA statistical analysis demonstrated significant group effect ($F_{3,36}$ = 85.04; P ≤ 0.0001) and time effect ($F_{3,36}$ = 9.279;

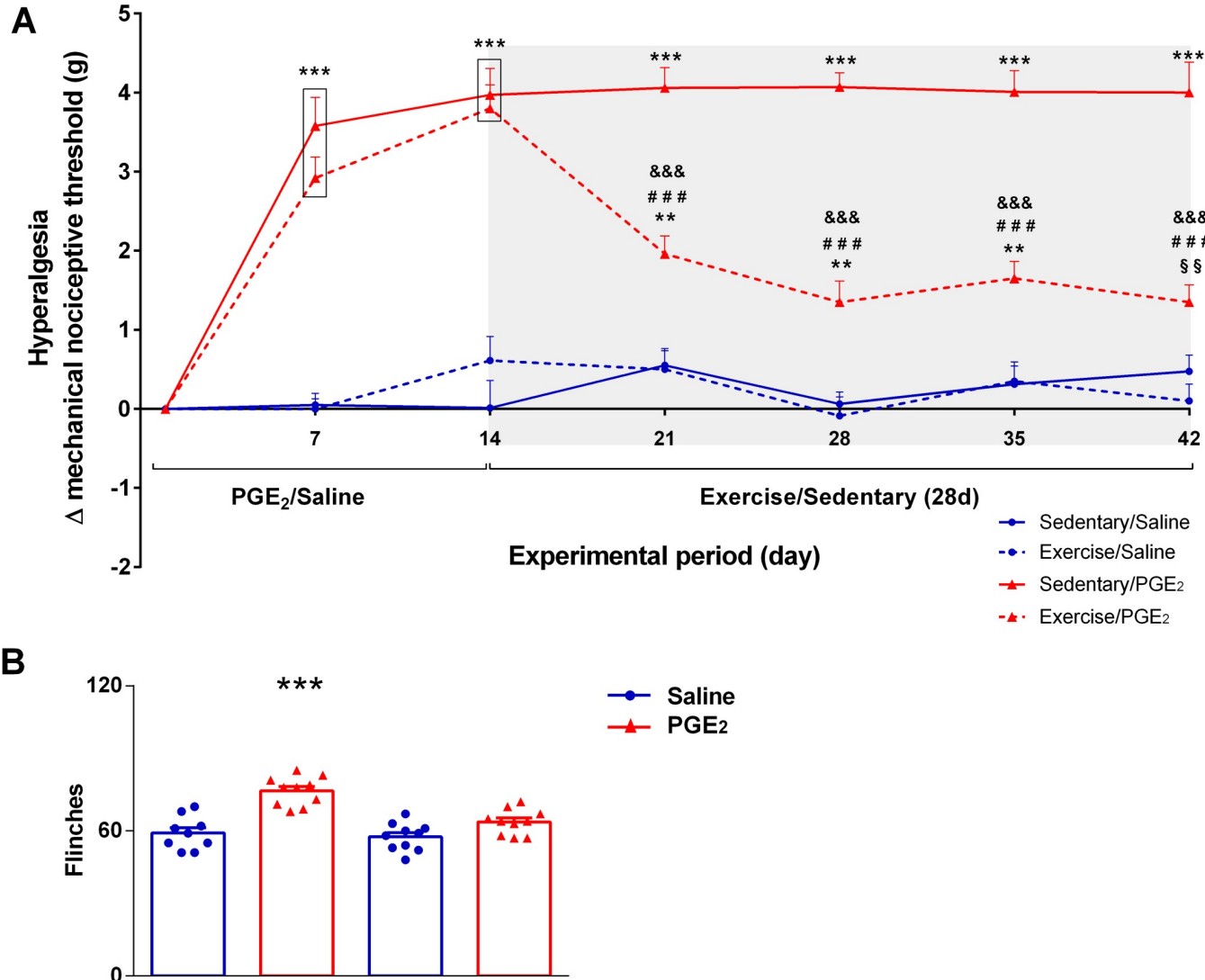

**Fig 2. Effect of exercise on persistent hyperalgesia. (A)** Effect of exercise on Δ mechanical nociceptive threshold. Mice from exercise/$PGE_2$ group showed significant reduction on hyperalgesia, while mice from sedentary/$PGE_2$ group remained with persistent hyperalgesia. Data are showed as mean ± SEM (N = 8–10 per group) (***p≤0.001, different from both sedentary/saline and exercise/saline groups; ###p≤0.001, different from sedentary/$PGE_2$ group; **p≤0.01, different from both sedentary/saline and exercise/saline groups; §§p≤0.01, different from exercise/saline group; &&&p ≤ 0.001, different from the timepoint 14d of the same group). Two-way ANOVA followed by Sidak posttest. The shaded area on the chart represents the period of the exercise. **(B)** Effect of exercise on nociceptive response to chemical stimuli. Sedentary mice injected with PGE2 (persistent hyperalgesia) showed more flinches (i.e. higher chemical hyperalgesia) compared to all the other groups. Exercised mice previously injected with PGE2 showed similar response compared to both saline groups. Data are showed as mean ± SEM (N = 9–10 per group; ***p≤0.001, different from all groups). One-way ANOVA followed by Bonferroni posttest.

P ≤ 0.0001). The *post hoc* Sidak multiple comparison test revealed that mice injected with PGE$_2$ showed higher hyperalgesia intensity compared to the control (saline) groups in time-points 7d and 14d (sedentary/PGE$_2$and exercise/PGE$_2$different from both sedentary/saline and exercise/saline; P ≤ 0.001). However, mice that performed exercise after persistent hyper-algesia induction (exercise/PGE$_2$) showed significant reduction in hyperalgesia intensity in the timepoints after beginning exercise (21d, 28d, 35d and 42d) compared to previous timepoints (7d and 14d; P ≤ 0.001). Moreover, mice from exercise/PGE$_2$ group showed lower hyperalge-sia intensity compared to mice from the sedentary/PGE$_2$ group in the timepoints after begin-ning exercise (21d, 28d, 35d and 42d; P ≤ 0.001). Furthermore, in the timepoint 42d, mice from exercise/PGE$_2$ group did not differ from sedentary/saline. On the other hand, mice from sedentary/PGE$_2$ group remained with higher hyperalgesia intensity compared to all other groups from the timepoint 21 until the end of the experimental period (P ≤ 0.001). Also, on timepoints 21d, 28d, 35d and 42d only exercise/PGE$_2$ differ statistically from the timepoint 14 in intragroup analysis (p ≤ 0.001).

**Fig 2B** shows the total number of flinches performed for each experimental group in the capsaicin test, representing the intensity of sensitivity to the chemical nociceptive stimuli. Statisti-cal analysis showed significant differences between groups ($F_{3,16}$ = 23.09; P ≤ 0.0001). The post hoc Bonferroni's multiple comparison test showed that mice from sedentary/PGE$_2$ group per-formed more flinches compared to the all other groups (sedentary/saline: 59.11 ± 2.282; seden-tary/PGE$_2$: 76.50 ± 1.875; exercise/saline: 57.50 ± 1.809; exercise/PGE$_2$: 63.70 ± 1.647; P ≤ 0.001).

## Experiment 2

**Even a short-term running wheel exercise reduced persistent hyperalgesia intensity.**
**Fig 3** shows the Experiment 2A experimental design and the mice activity on running wheel. Two-way ANOVA statistical analysis demonstrated significant time effect ($F_{13,234}$ = 77.67; P ≤ 0.0001) and no group effect ($F_{13,234}$ = 1.306; P = 0.0220), considering the average distance run on each day over the 28 days of running wheel free access. The *post hoc* Sidak multiple comparison test revealed no significant differences between groups. Moreover, considering the average distance run per day on the total period (**Fig 3C**), unpaired T test showed no sig-nificant differences between groups (PGE$_2$/Exercise 14d-28d: 4.107 ± 0.4130 km/day; saline/Exercise 14d-28d: 4.909 ± 0.3920 km/day; P = 0.1762; t = 1.408 df = 18). Same can be seen for area under the curve (**Fig 3D**) where unpaired T test showed no significant differences between groups (saline/Exercise 14d-28d: 53.72 ± 5.397; PGE$_2$/Exercise 14d-28d: 65.38 ± 5.690; P = 0.1544; t = 1.487 df = 18).

**Fig 4** shows the Experiment 2B experimental design and the mice activity on running wheel. Two-way ANOVA statistical analysis demonstrated significant time effect ($F_{13,221}$ = 36.41; P ≤ 0.0001) and no group effect ($F_{13,221}$ = 0.9880; P = 0.4640), considering the average distance run on each day over the 28 days of running wheel free access. The *post hoc* Sidak multiple comparison test revealed no significant differences between groups. Furthermore, considering the average distance run per day on the total period (**Fig 4C**), unpaired T test showed no significant differences between groups (PGE$_2$/Exercise 28d-42d: 4.008 ± 0.4216 km/day; saline/Exercise 28d-42d: 4.265 ± 0.3648 km/day; P = 0.6537; t = 0.4566 df = 17). Same can be seen for area under the curve (**Fig 4D**) where unpaired T test showed no significant dif-ferences between groups (saline/Exercise 28d-42d: 55.66 ± 4.757; PGE$_2$/Exercise 28d-42d: 52.36 ± 5.556; P = 0.6614; t = 0.4458 df = 17).

**Fig 5A** shows the variation of the mechanical nociceptive threshold in grams (Δ mechanical nociceptive threshold), representing the hyperalgesia intensity for all groups from Experiment 2. Two-way ANOVA statistical analysis demonstrated significant group effect ($F_{3,31}$ = 87.96;

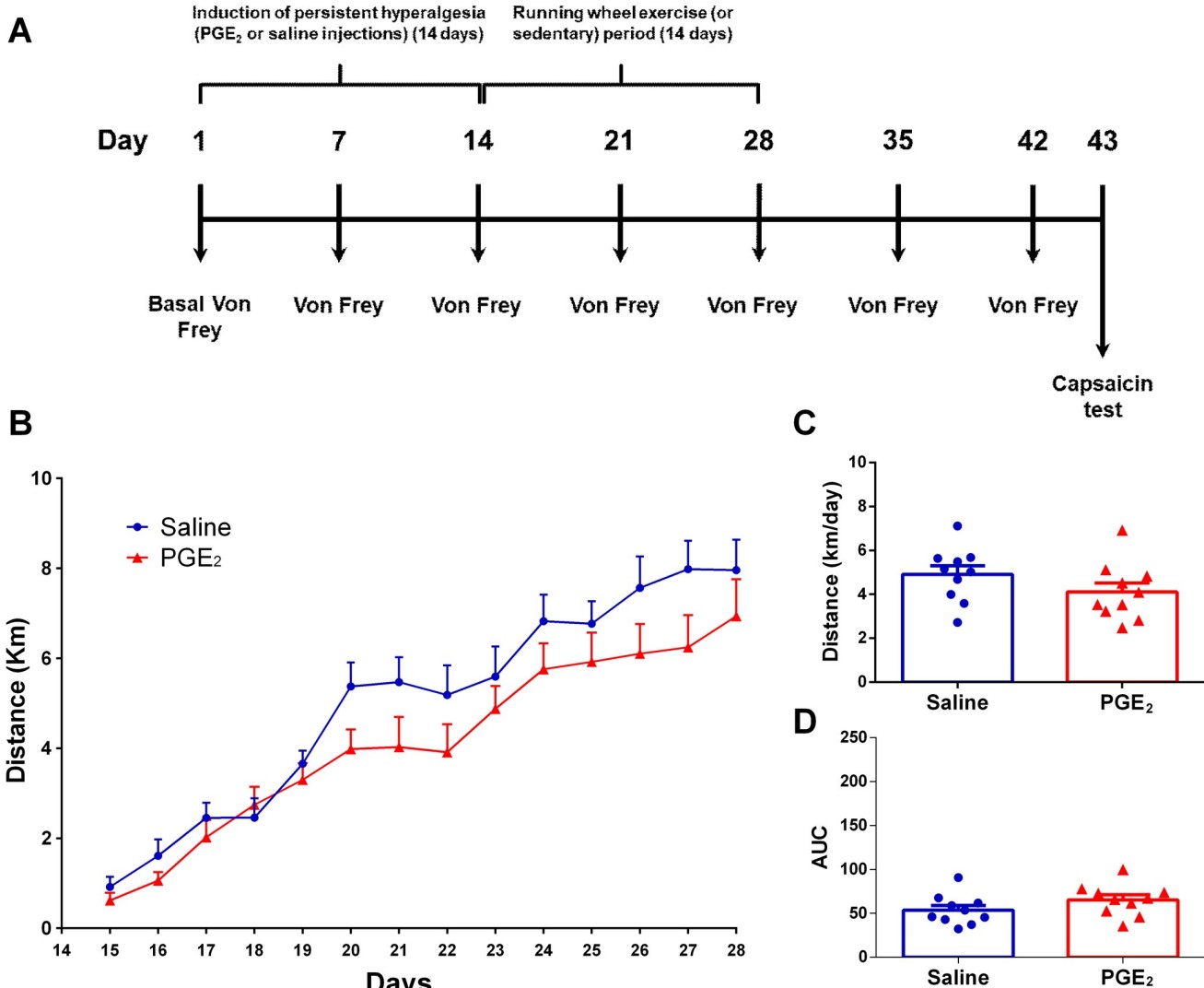

**Fig 3. Mice activity on running wheel. (A)** Graphical diagram of the experimental design. In this experiment mice start running from 14d to 28d. **(B)** Average distance (km) traveled on each day over 14 days of running wheel free access. **(C)** Average distance (km) traveled per day on all 14 days of running wheel free access. **(D)** Area under the curve (AUC) for each group calculated from (A) data. There were no significant differences between the groups. Data are showed as mean ± SEM (N = 10 per group). Two-way ANOVA followed by Sidak posttests and multiple t tests (B); Unpaired t test (C) and (D).

$P \leq 0.0001$) and time effect ($F_{6,186} = 56.82$; $P \leq 0.0001$), considering the average distance run on each day over the 28 days of running wheel free access. The post hoc Sidak multiple comparison test revealed that mice injected with $PGE_2$ ($PGE_2$/Exercise 14d-28d and $PGE_2$/Exercise 28d-42d) showed higher hyperalgesia intensity compared to the control (saline) groups (Saline/Exercise 14d-28d and Saline/Exercise 28d-42d) in timepoints 7d and 14d($P \leq 0.001$). Also, mice that performed exercise after persistent hyperalgesia induction from timepoint 14d to 28d ($PGE_2$/Exercise 14d-28d group–blue shaded area in the [Fig 5A]) showed significant reduction in hyperalgesia intensity in all timepoints after beginning exercise (21d, 28d, 35d and 42d) compared to previous timepoints (7d and 14d; $P \leq 0.001$) and showed significant lower hyperalgesia intensity compared to mice that would starting running later ($PGE_2$/Exercise 28d-42d group) ($P \leq 0.001$). However, they remained statistically different from all saline

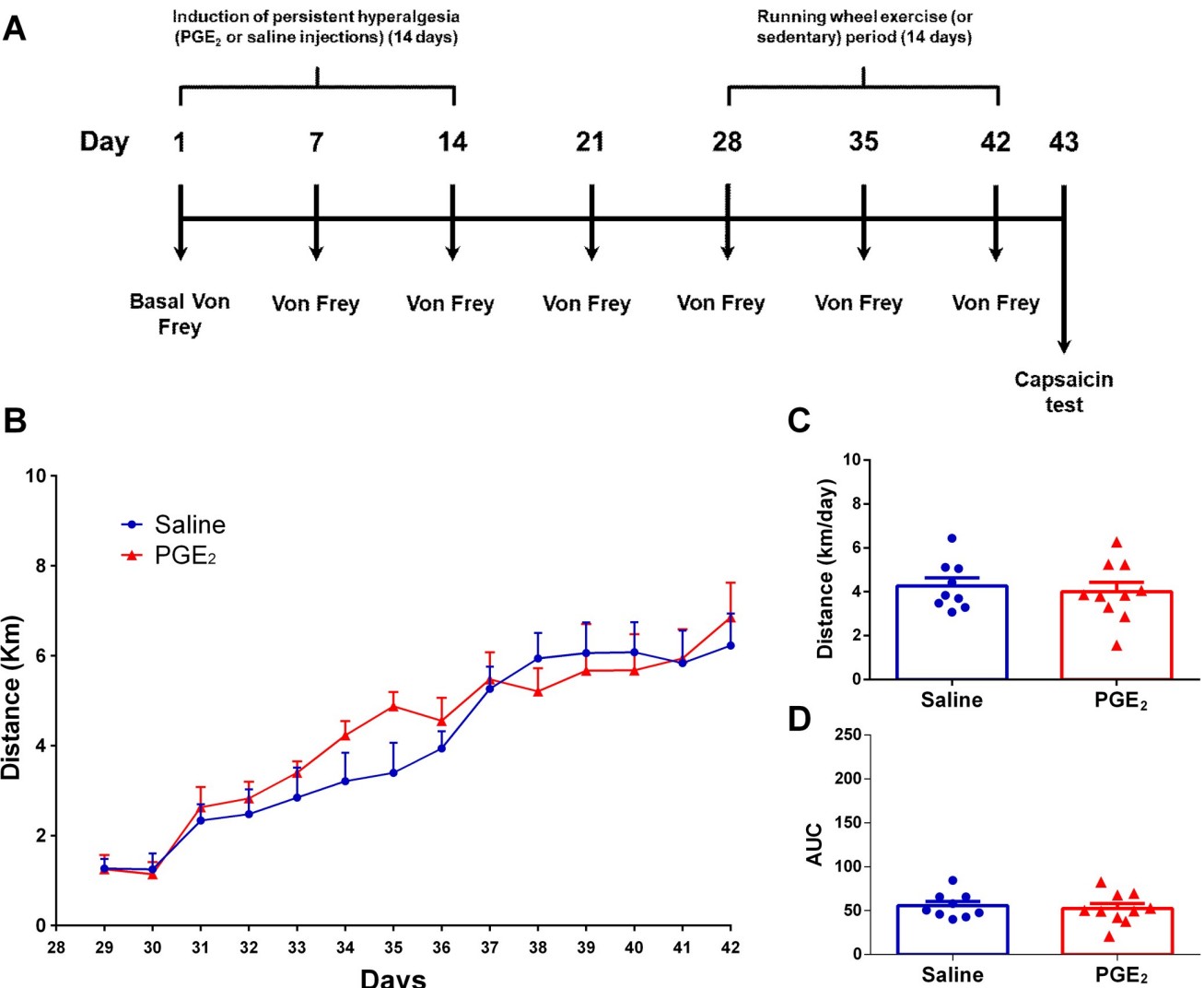

**Fig 4. Mice activity on running wheel. (A)** Graphical diagram of the experimental design. In this experiment mice start running from 28d to 42d. **(B)** Average distance (km) traveled on each day over 14 days of running wheel free access. **(C)** Average distance (km) traveled per day on all 14 days of running wheel free access. **(D)** Area under the curve (AUC) for each group calculated from (A) data. There were no significant differences between the groups. Data are showed as mean ± SEM (N = 10 per group). Two-way ANOVA followed by Sidak posttests (B); Unpaired t test (C) and (D).

groups (P ≤ 0.001). Moreover, mice that performed exercise after persistent hyperalgesia induction from timepoint 28d to 42d ((PGE$_2$/Exercise 28d-42d group–red shaded area in the **Fig 5A**) showed significant reduction in hyperalgesia intensity in all timepoints after beginning exercise (35d and 42d) compared to previous timepoints (7d, 14d, 21d and 28d,; P ≤ 0.001) and became statistically equal to the mice that ran from 14d to 28d. Finally, they remained statistically different from all saline groups (P ≤ 0.001). **Fig 5B** shows the total number of flinches performed for each experimental group in the capsaicin test, representing the intensity of sensitivity to the chemical nociceptive stimuli. Statistical analysis showed no significant differences between groups ($F_{3,32}$ = 1.072; P = 0.3748). The post hoc Bonferroni's multiple comparison test also showed no differences between groups (Saline/Exercise 14d-28d: 56.67 ± 2.088; PGE$_2$/Exercise 14d-28d: 60.56 ± 2.534; Saline/Exercise 28d-42d: 55.00 ± 1.787; PGE$_2$/Exercise 28d-42d: 58.11 ± 2.590).

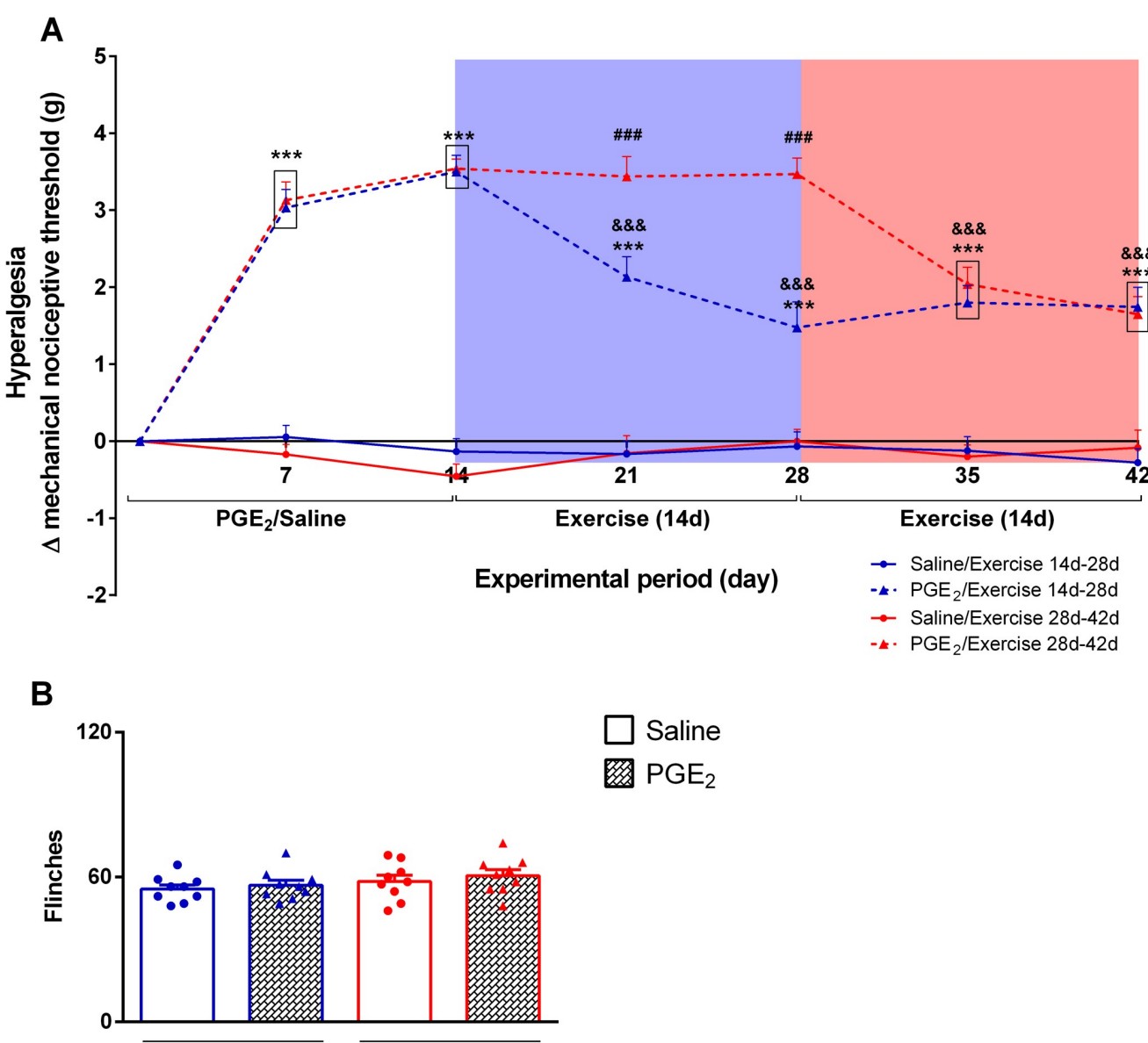

**Fig 5. Effect of exercise on persistent hyperalgesia. (A)** Effect of exercise on Δ mechanical nociceptive threshold. Mice from exercisegroup showed significant reduction on hyperalgesia after start running. Blue shaded area indicates the period that the groups represented by blue lines performed exercise and the red shaded area indicates the period that the groups represented by red lines performed exercise. Data are showed as mean ± SEM (N = 9–10) (***p≤0.001, different from both saline groups; ###p ≤ 0.001, different from PGE2/Exercise 14d-28d group; &&&p ≤ 0.001, different from the timepoints 7d and 14d of the same group). Two-way ANOVA followed by Bonferroni posttest. **(B)** Effect of exercise on nociceptive response to chemical stimuli. There is no statistical difference between groups. Data are showed as mean ± SEM (N = 9 per group). One-way ANOVA followed by Bonferroni posttest.

### Experiment 3

**Repetitive prostaglandin E2 injections reduced mice activity in the running wheel.** **Fig 6** shows the Experiment 3 experimental design and the mice activity on running wheel. Two-way ANOVA statistical analysis demonstrated significant time effect ($F_{1,18}$ = 10.01; P ≤ 0.0001) and group effect ($F_{1,18}$ = 6.288; P ≤ 0.0220), considering the average distance run on each day over the 28 days of running wheel free access (**Fig 6B**). The *post hoc* Sidak multiple comparison test revealed that mice injected with PGE2 show less activity on timepoints 22d to

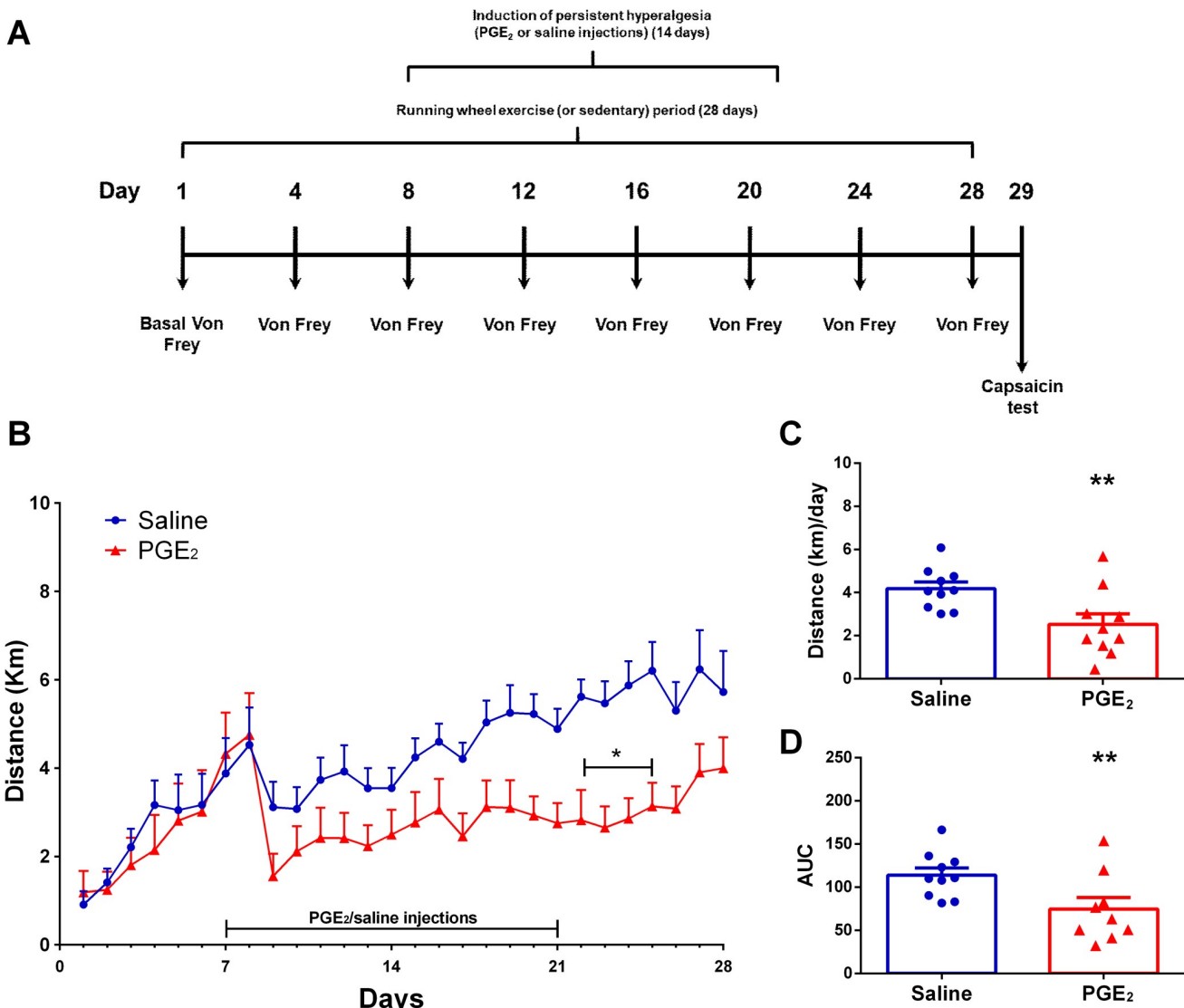

**Fig 6. Mice activity on running wheel. (A)** Graphical diagram of the experimental design. **(B)** Average distance (km) traveled on each day over 28 days of running wheel free access (*$p \leq 0.05$ different from saline group at same timepoint; **$p \leq 0.01$ different from saline group at same timepoint). **(C)** Average distance (km) traveled per day on all 28 days of running wheel free access (**$p \leq 0.01$ different from saline group). **(D)** Area under the curve (AUC) for each group calculated from (A) data (**$p \leq 0.01$ different from saline group). Data are showed as mean ± SEM (N = 10 per group). Two-way ANOVA followed by Sidak posttests (A); Unpaired t test (B) and (C).

25d ($P \leq 0.05$). Moreover, considering the average distance run per day on the total period (**Fig 6C**), unpaired T test showed significant differences between groups (exercise/PGE$_2$: 2.524 ± 0.4923 km/day; exercise/saline: 4.183 ± 0.3012 km/day; P = 0.0101). Same can be seen for area under the curve (**Fig 6D**) where unpaired T test showed significant differences between groups (exercise/PGE$_2$: 113.8 ± 8.290; exercise/saline: 74.55 ± 13.21; P = 0.0198).

**Running wheel exercise prevented persistent hyperalgesiadevelopment.** **Fig 7A** shows the variation of the mechanical nociceptive threshold in grams (Δ mechanical nociceptive threshold), representing the hyperalgesia intensity. Two-way ANOVA statistical analysis demonstrated significant group effect ($F_{3,36} = 15.10$; $P < 0.0001$) and time effect ($F_{3,36} = 26.82$; $P \leq 0{,}0001$). The *post hoc* Sidak multiple comparison test revealed that mice performing

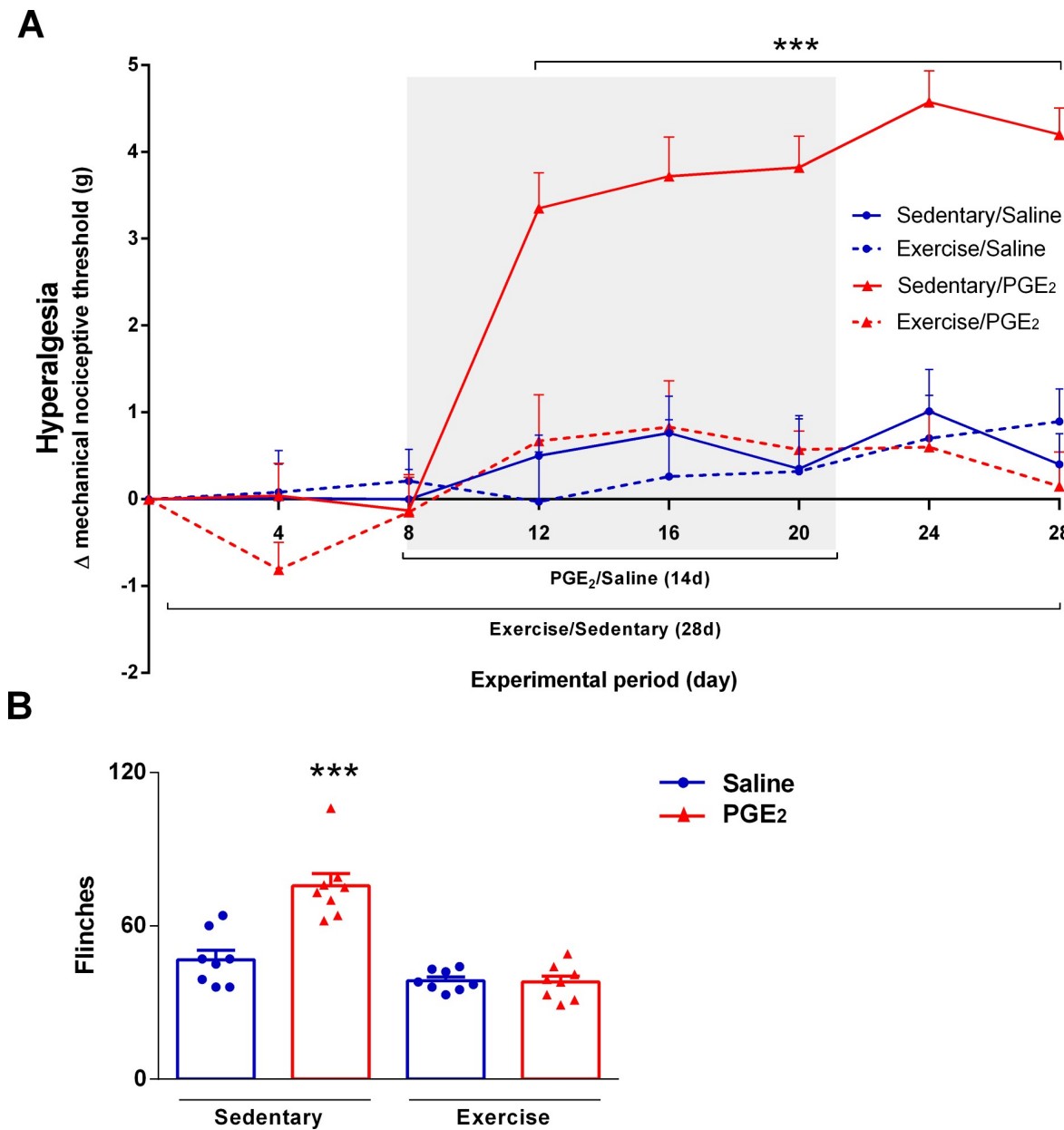

**Fig 7. Effect of exercise on persistent hyperalgesia development.** (A) Preventive effect of physical exercise on Δ mechanical nociceptive threshold during persistent hyperalgesia induction. Mice fromexercise/PGE₂groupdid not present persistent hyperalgesia. Data are presented as mean ± SEM (N = 10 per group). (***p ≤ 0.001, different from all other groups at the same timepoint). Two-way ANOVA followed by Bonferroni posttest. The shaded area on the chart represents the period of the persistent hyperalgesia induction by repetitive PGE₂ injections. (B) Effect of exercise on nociceptive response to chemical stimuli. Sedentary mice injected with PGE₂(persistent hyperalgesia) showed more flinches (i.e. higher chemical hyperalgesia) compared to all the other groups. Mice performing exercise simultaneously to PGE₂ injections showed similar response compared to both saline groups. Data are showed as mean ± SEM (N = 8 per group; ***p≤0.001, different from all groups). One-way ANOVA followed by Bonferroni posttest.

exercise simultaneously to repetitive inflammatory stimulus injectiondid not present persistent hyperalgesia development. Only mice from the sedentary/PGE₂ group presented significant persistent hyperalgesia compared to all the other groups (P ≤ 0.001).

**Fig 7B** shows the total number of flinches performed for each experimental group in the capsaicin test, representing the intensity of sensitivity to the chemical nociceptive stimuli.

Statistical analysis showed significant differences between groups ($F_{3,16}$ = 28.20; P $\leq$ 0.0001). The post hoc Bonferroni's multiple comparison test showed that mice from sedentary/PGE$_2$ group performed more flinches compared to the all other groups (sedentary/saline: 46.75 ± 3.702; sedentary/PGE$_2$: 75.63 ± 4.807; exercise/saline: 38.50 ± 1.427; exercise/PGE$_2$: 38.00 ± 2.398; P $\leq$ 0.001).

## Discussion

The present study aimed to investigate the therapeutic and preventive effect of the running wheel exercise on the persistent hyperalgesia induced by repetitive inflammatory stimulus (PGE$_2$) in mice. Our results demonstrated that voluntary running wheel exercise can reduce persistent hyperalgesia intensity (**experiments 1 and 2; Figs 2 and 5**). In addition, experiment 2 demonstrated that the antihyperalgesic effect of physical exercise was long-lasting (**experiment 2A; Fig 5**), and therapeutic effect of exercise for persistent hyperalgesia was maintained even with the late onset of the intervention (**experiment 2B; Fig 5**). Also, our results demonstrated a robust preventive effect of physical exercise for pain chronification, once it prevented the development of persistent hyperalgesia (**experiment 3; Fig 7**).

To the best of our knowledge, this is the first study showing the therapeutic and preventive effects of physical exercise on a mouse model of persistent hyperalgesia induced by repetitive inflammatory stimulus. It is well established that, in this model, mice show persistent hyperalgesia up to 30 days after ending the PGE$_2$ injections [24] and this data was replicated in the present study. This is an appropriate model to study susceptibility and resilience to pain chronification [26, 27], allowing us to assess the central hypothesis of the present study that physical exercise promotes resilience to chronic pain and can be used as a non-pharmacological therapeutic and preventive tool for this condition.

The great complexity in the development and maintenance of chronic pain, as well as the difficulty in performing clinical trials with significant sample, makes rare studies showing the efficacy of physical exercise as a non-pharmacological intervention for the treatment and prevention of this condition ([8–14] for review see [28]). Moreover, placebo effect is a confounding factor on clinical trials studies [29], which is bypassed in animal models, like in the present study. In addition, most of the studies in animals used models of neuropathic pain [15–21], differently of the present study, where we used a chronic pain model that employs original inflammatory stimulus, mimicking the main types of pain in the clinical setting [22].

It is important to highlight some studies that corroborate the therapeutic effects of physical exercise observed in experiment 1 and 2. Cobianchi *et al.* [30], for example, showed in mice that short-lasting treadmill running reduced nerve injury-induced chronic neuropatic pain and facilitates the regenerative processes of the injured nerve. In addition, Whitehead *et al.* [18] demonstrated that regular voluntary exercise can reverse sciatic nerve ligation-induced hyperalgesia in rats. Kuphal *et al.* [31], using a swimming forced exercise approach, also demonstrated reduced hyperalgesia in formalin and sciatic nerve ligation animal models. Similarly, Bobinski *et al.* [32], using another forced physical exercise approach, showed that two weeks of treadmill exercise reversed sciatic ligature-induced neuropathic pain. More recently, Bobinski *et al.* [19] showed that 2 weeks of treadmill exercise improves mechanical hyperalgesia induced by peripheral nerve injury.

Although the present study did not aim to investigate the molecular mechanisms of the antihyperalgesic effect of physical exercise, other researchers have adopted this approach and may explain this effect. Bobinski *et al.* [19], for example, showed that physical exercise-induced analgesia is mediated by IL-4, an anti-inflammatory cytokine. In fact, physical exercise seems to upregulate anti-inflammatory cytokines in favor to pro-inflammatory cytokines, leading to

an analgesic physiological environment [19, 33–36]. Also, corroborating this idea, physical exercise increases the IL-10/TNF-α ratio, an important condition for an anti-inflammatory profile [37]. This highlights the impact of the present study where we showed that physical exercise can reduce and prevent hyperalgesia induced by a model that does not induce the release of inflammatory cytokines, once we injected the final inflammatory mediator $PGE_2$ [23, 24]. Based on this, we can hypothesize that, in this study, exercised mice show reduced endogenous $PGE_2$, because of the exercise-induced anti-inflammatory profile, and great $PGE_2$ degradation induced by increased anti-inflammatory cytokines [38]. Taken together, these aftereffects can be used as a theoretical basis to explain the therapeutic and preventive effects of physical exercise on $PGE_2$-induced chronic pain, observed in the present study.

Besides the abovementioned effects on immune system, physical exercise triggers several alterations in the whole body that could be associated with the antihyperalgesic effect observed in the present study. In this context, it is well known the role of serotonergic and opioidergic systems as mediators of the exercise-induced analgesia [16, 32, 39–41]. Lima *et al.* [39], for example, showed that voluntary running wheel exercise can reverse the hyperalgesia induced by a mouse model of activity-induced pain, also demonstrating–after systemic administration of naloxone (opioid receptor antagonist)–that this exercise-induced analgesia is opioid dependent. Similarly, Stagg *et al.* [16] showed that the opioid system, both systemically and in the brainstem, are involved in the physical exercise-induced analgesia. The endogenous opioid system mediating the antihyperalgesic effect of exercise has also been suggested in a study of chemical-induced pain [40], as well as in another of chronic muscle pain [41]. Based on this, we can hypothesize that, in our model, the exercise can simultaneously activate the serotonergic and opioidergic systems, reversing and preventing the $PGE_2$-induced chronic pain.

In an interesting experiment, Almeida and colleagues [42] corroborated our data and showed that forced exercise can reverse chronic hyperalgesia induced by partial nerve injury. They also showed that exercise reversed nerve injury-induced BDNF upregulation and glial hyperactivity on dorsal root ganglion (DRG), which can also be used as background to explain the mechanism of the exercise-induced analgesia showed in the present study. In this context, it is important to mention that DRG glial activation is required for the $PGE_2$-induced hyperalgesia [43, 44], which suggest that the analgesia induced by exercise observed in the present study could be mediated by a DRG glial mechanism.

In addition, it is important to mention the well-known effect of physical exercise ameliorating oxidative stress parameters. In fact, several studies have already shown exercise improving oxidative stress and antioxidant parameters [45–47]. Also, it was already demonstrated the effect of oxidative stress on chronic pain induction and establishment [48–50]. Based on this, we can hypothesize that, in our study, the low levels of oxidative stress on exercised mice ameliorated the $PGE_2$-induced chronic pain, also triggering a protective physiological environment which prevented the $PGE_2$-induced chronic pain. In this context, it is interesting to mention the role of the cannabinoid system, and its ligands, on reducing oxidative stress parameters [51]. In fact, the direct effect of physical exercise on the endocannabinoid system can also explain the exercise-induced analgesia observed in the present study, once the endocannabinoid system has emerged as the main mechanisms for how exercise benefits the whole organism and how it reduces and controls pain [52].

As demonstrated by the above-mentioned studies, investigating molecular mechanisms associated with the antihyperalgesic effects of physical exercise is very interesting and important for the field. However, our group believes that exercise is a behavior that engages the entire body and its effects are systemic and multiple, inducing biological adaptations in several tissues such as nervous, musculoskeletal, cardiovascular, respiratory and immune systems [36, 53, 54]. Thus, it is plausible to consider that multiple biological mechanisms underlying

exercise-induced analgesia act synergistically, involving countless molecular agents. So, we believe that the precise understanding of molecular mechanisms behind physical exercise anti-hyperalgesic effects is less relevant than that related to drugs studies, since drugs may have important side effects that need to be prevented while exercise is a natural behavior regulated by very secure physiological feedback loops.

It is important to emphasize that most of the animal studies above-mentioned used forced physical exercise. In the present study we used voluntary physical exercise, once our research group believes that voluntary physical exercise better mimics aspects of human physical activity: evolutionarily forged to run purposefully [55]. In conclusion, our results showed that physical exercise can be used as a therapeutic and preventive approach for chronic pain. Based on our results, we may speculate that exercise can be associated with other therapeutic modalities, including pharmacotherapy, reducing drug doses and their harmfulside effects. Moreover, the preventive effect of physical exercise showed in the present study strongly supports its recommendation to promote resilience to the development of chronic pain conditions, which is obviously inappropriate and impractical with drug prescription.

## Author Contributions

**Conceptualization:** Cesar Renato Sartori, Carlos Amilcar Parada.

**Data curation:** Cesar Renato Sartori, Ivan José Magayewski Bonet.

**Formal analysis:** Cesar Renato Sartori, Marco Pagliusi, Jr, Ivan José Magayewski Bonet, Claudia Herrera Tambeli, Carlos Amilcar Parada.

**Funding acquisition:** Cesar Renato Sartori, Carlos Amilcar Parada.

**Methodology:** Cesar Renato Sartori, Ivan José Magayewski Bonet.

**Project administration:** Cesar Renato Sartori.

**Supervision:** Carlos Amilcar Parada.

**Writing – original draft:** Cesar Renato Sartori, Marco Pagliusi, Jr, Claudia Herrera Tambeli, Carlos Amilcar Parada.

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
