## [Decision Letter · Decision Letter 0]

8 Apr 2020

PONE-D-20-03419

Running wheel exercise induces therapeutic and preventive effects on inflammatory stimulus-induced persistent hyperalgesia in mice.

PLOS ONE

Dear Dr. Sartori,

Thank you for submitting your manuscript to PLOS ONE. After careful consideration, we feel that it has merit but does not fully meet PLOS ONE’s publication criteria as it currently stands. Therefore, we invite you to submit a revised version of the manuscript that addresses the points raised during the review process.

We would appreciate receiving your revised manuscript by May 23 2020 11:59PM. To enhance the reproducibility of your results, we recommend that if applicable you deposit your laboratory protocols in protocols.io, where a protocol can be assigned its own identifier (DOI) such that it can be cited independently in the future. For instructions see: http://journals.plos.org/plosone/s/submission-guidelines#loc-laboratory-protocols

We look forward to receiving your revised manuscript.

Kind regards,

Michael Costigan

Academic Editor

PLOS ONE

Journal Requirements:

2. To comply with PLOS ONE submissions requirements, please provide methods of sacrifice in the Methods section of your manuscript.

Reviewers' comments:

Reviewer's Responses to Questions

**Comments to the Author**

1. Is the manuscript technically sound, and do the data support the conclusions?

Reviewer #1: Partly

Reviewer #2: Partly

2. Has the statistical analysis been performed appropriately and rigorously? 

Reviewer #1: Yes

Reviewer #2: No

3. Have the authors made all data underlying the findings in their manuscript fully available?

Reviewer #1: Yes

Reviewer #2: Yes

4. Is the manuscript presented in an intelligible fashion and written in standard English?

Reviewer #1: Yes

Reviewer #2: No

5. Review Comments to the Author

Reviewer #1: The authors show the analgesic potential of running wheel exercise in the persistent hyperalgesia induced by repetitive inflammatory stimulus. The scientific logic flow to design and perform the experiments is clear. The manuscript is well written but a few points need to be addressed.

1. It is necessary a better description of the methodology. ¿what kind of wheels are you using (metallic or plastic, which size?), the animals shared the wheel or they have one wheel each, are the animals grouped in cages or isolated?

2. Electronic von Frey test is described as nociceptive test, but you are using it to measure mechanical hyperalgesia. Please do not mix both concepts. Page 5, lines 159 and 162.

3. As suggested by Cornier et al. 2017 and Pitcher et al. 2017, and even by the authors in the discussion section, the role of the endogenous opioid could be fundamental in the effect showed in the manuscript. Although was pointed by the authors that differently to pharmacological approaches the molecular mechanisms are less relevant in the exercise, I believe that in this publication is necessary to clarify the molecular mechanism to avoid confusions by methodological interferences as changes in the paw skin by the constant use of the wheels. For this reason, would be interesting to see if the analgesia produced by the wheels is avoided by the opioid antagonist naloxone and check the plasma corticosteroids.

4. The authors should homogenise the Y axes in the different figures to make them easily comparables. As example, figure 2B show flinches from 40 to 90 and Figure 7B show flinches from 0 to 120.

Reviewer #2: ID: PONE-D-20-03419

These investigators injected the inflammatory mediator PGE2 or vehicle intraplantarly into the hindpaw of mice on a daily basis for 14 days and then at day 14 the injections were stopped and running wheels (or no wheel controls) were introduced to the mouse cages for 28 days. Mice were tested weekly for hindpaw von Frey hyperalgesia and at 7 and 14 days of PGE2 injection the mice had hyperalgesia. Having free access to the exercise wheel partially reduce hyperalgesia after 7, 14, 21, and 28 days of exercise. There were no differences in daily running distance between the vehicle and PGE2 injected mice. Capsaicin intraplantar injection caused increased flinching in no wheel/no PGE treatment group compared to vehicle treated or PGE2/wheel treated mice after 28 days of wheel access. These results indicate an analgesic effect of wheel exercise on PGE2 induced chronic von Frey hyperalgesia and capsaicin induced flinching, but not on daily distance running.

Another experiment started with a 14 day course of daily PGE2 or vehicle intraplantar injections, then 14 days of running wheel access or no wheel access. There was no difference in the distance run/day between PGE2 and vehicle. Von Frey hindpaw threshold testing was performed weekly during the entire time period and for 14 days after the wheel access ended. The PGE2 mice developed von Frey hyperalgesia by day 7 of injections and wheel exercise starting at the end of the PGE2 treatment partially reversed this hyperalgesia and this effect persisted for 14 days after stopping wheel exercise. Capsaicin intraplantar injection caused no increased flinching in PGE2/wheel treated mice 14 days after wheel access.

Another experiment started with a 14 day course of daily PGE2 or vehicle intraplantar injections, then 14 days of no treatment, then 14 days of running wheel access or no wheel access. There was no difference in the distance run/day between PGE2 and vehicle. The PGE2 mice developed von Frey hyperalgesia by day 7 of injections and wheel exercise starting at 14 days after the end of the PGE2 treatment partially reversed this hyperalgesia. Capsaicin intraplantar injection caused no increased flinching PGE2/wheel treated mice after 14 days of wheel access.

Another experiment provided the mice with 28 days wheel access or no wheel access, starting at day 1, then after 8 days of wheel or no wheel access PGE2 or vehicle daily intraplantar injections were started for 14 days. The PGE2 mice did display a reduction in daily wheel running distance relative to the vehicle group. The PGE2/no wheel mice developed von Frey hyperalgesia by day 7 of PGE2 injections, but the PGE2/ wheel exercise group did not develop hyperalgesia. Capsaicin intraplantar injection caused no increased flinching PGE2/wheel treated mice after 14 days of wheel access.

Theses results are novel and the experiments seem to be well performed but there are several issues.

Major Concerns:

1. Repeated testing over time within the same groups of mice need to be analyzed with repeated measures 2 way ANOVA with Sidak post hoc testing.

2 The introduction and discussion need extensive revision with an English speaking editor.

3. There are many paragraphs in the introduction and discussion that are tangential or do not add to hypothesis or data, would recommend deleting lines 45-59, 76-78, 384-393, 418-422, 459-470.

4. There is no discussion regarding the mechanisms by which repeated PGE2 injections cause long lasting hyperalgesia. Dr Parada coauthored many papers on this topic with Dr Ferreira and it is surprising there is no discussion on how PGE2 injections cause long acting hyperalgesia by up-regulating the Nav1.8 sodium channels in the DRG, that PGE2 hyperalgesia is mediated by nonpeptidergic small sensory neurons that normally do not contribute to von Frey thresholds, and most importantly, that PGE2 induced mechanical hyperalgesia is mediated by DRG glutamate release acting on the NMDA receptor on DRG satellite cells. DRG satellite cells are immune cells capable of releasing proinflammatory cytokines and growth factors that may mediate chronic PGE2 sensitization.

5. The discussion of the prior literature on exercise analgesic mechanisms in rodent pain models is superficial and incomplete. More discussion of exercise suppression of pronociceptive immune responses is required and this should be tied into the crucial role of DRG satellite immune cells in PGE2 chronic hyperalgesia. The following papers should be included and reviewed; Lopez- Alvarez et al, Pain 2015: 156: 1812-25; Almeida et al, Pain 2015:156:504- 513; Cobianchi et al, Neuroscience 2010: 168: 273-287; Shi et al, Anesthesiology 2018: 129: 557-75.

6. This is a descriptive paper with no mechanism or hypothesis of a mechanism for the analgesic effects of exercise.

Minor concerns:

1. Need to make clear in methods if mice were housed in individual cages and if they had the wheel in their cage 24 hours/day.

2. There is no information on the running wheels, vendor, or computerized data collection methods for measuring distance run/day?

3. Why did the authors inject the same amount and volume of PGE2 in mice as they previously used in rats, shouldn’t smaller doses and volumes be used in much smaller animals, please add to the discussion in a section on the limitations of the study.

4. All figure legends need to be on one page.

6. PLOS authors have the option to publish the peer review history of their article (what does this mean?). If published, this will include your full peer review and any attached files.

Reviewer #1: No

Reviewer #2: No

---

## [Author Response · Author response to Decision Letter 0]

5 Jun 2020

Response to Reviewer #1

First, we would like to thank the contributions made by the reviewer. Below are our comments, point-by-point, to the comments and questions raised. We are at your disposal for any additional clarification that may be required.

Reviewer #1: The authors show the analgesic potential of running wheel exercise in the persistent hyperalgesia induced by repetitive inflammatory stimulus. The scientific logic flow to design and perform the experiments is clear. The manuscript is well written but a few points need to be addressed.

1. It is necessary a better description of the methodology. ¿what kind of wheels are you using (metallic or plastic, which size?), the animals shared the wheel or they have one wheel each, are the animals grouped in cages or isolated?

In the revised manuscript (“Voluntary running wheel exercise” section, Methods) we included a better description of the methodology. As requested by the reviewer, we included the informations that the wheels are plastic with 9.2 cm in diameter, as well as, manufacturer informations. Also, after starting the experimental protocol all mice remained single housed with one wheel each. 

2. Electronic von Frey test is described as nociceptive test, but you are using it to measure mechanical hyperalgesia. Please do not mix both concepts. Page 5, lines 159 and 162.

The authors understand that both concepts (nociceptive test and mechanical hyperalgesia) should not be mixed and confused. It is important to mention that, in the present study, we choose to show the mechanical nociceptive test results as the variation of the mechanical nociceptive threshold, which we call mechanical hyperalgesia. This variation is calculated by subtracting the average of three values observed in the basal von Frey test from those of the average of three values obtained in each of all following tests performed. Aiming to avoid misunderstanding, we rewrote some sentences of the “Electronic von Frey” section, Methods. 

3. As suggested by Cornier et al. 2017 and Pitcher et al. 2017, and even by the authors in the discussion section, the role of the endogenous opioid could be fundamental in the effect showed in the manuscript. Although was pointed by the authors that differently to pharmacological approaches the molecular mechanisms are less relevant in the exercise, I believe that in this publication is necessary to clarify the molecular mechanism to avoid confusions by methodological interferences as changes in the paw skin by the constant use of the wheels. For this reason, would be interesting to see if the analgesia produced by the wheels is avoided by the opioid antagonist naloxone and check the plasma corticosteroids.

The authors considered pertinent the experiment suggested by the reviewer. However, this approach has already been extensively tested by other groups. Lima et al (2017), for example, showed that running wheel exercise-induced analgesia is avoided by the opioid antagonist naloxone. Mazzardo-Martins et al (2010) showed that naloxone reversed forced exercise-induced analgesia in a muscle pain model, similarly as observed by Shankarappa et al (2011) in a diabetes-induced neuropatic pain model. Also, Bement and Sluka (2005) showed, using von Frey as nociceptive test, that naloxone reversed the low intensity exercise-induced analgesia in the chronic muscle pain mice model. Although the above-mentioned studies were performed in different chronic pain models from that used in the present study, we considered redundant assess the involvement of the opioid system, mainly using naloxone, once this system is well known engaged in the exercise-induced analgesia context. Finally, the effects of exercise in the plasma corticosteroids concentration has also been extensively assed on previous studies (Lynch et al, 2019; Radahmadi et al, 2015; Girard and Garland, 2002; Coleman et al, 1998). Because of this well-established mechanism that we prefer to consider a whole-body effect of the exercise, as we discussed in the manuscript. However, considering the reviewer commentary, in the revised manuscript we extended the discussion about the mechanism involved in the exercise-induced analgesia (“Discussion” section; lines 394 to 405).

4. The authors should homogenise the Y axes in the different figures to make them easily comparables. As example, figure 2B show flinches from 40 to 90 and Figure 7B show flinches from 0 to 120

The authors agree with the reviewer and we homogenized the Y axes in the figures of the revised manuscript. 

-----

Response to Reviewer #2

First, we would like to thank the contributions made by the reviewer. Below are our comments, point-by-point, to the comments and questions raised. We are at your disposal for any additional clarification that may be required.

Reviewer #2: These investigators injected the inflammatory mediator PGE2 or vehicle intraplantarly into the hindpaw of mice on a daily basis for 14 days and then at day 14 the injections were stopped and running wheels (or no wheel controls) were introduced to the mouse cages for 28 days. Mice were tested weekly for hindpaw von Frey hyperalgesia and at 7 and 14 days of PGE2 injection the mice had hyperalgesia. Having free access to the exercise wheel partially reduce hyperalgesia after 7, 14, 21, and 28 days of exercise. There were no differences in daily running distance between the vehicle and PGE2 injected mice. Capsaicin intraplantar injection caused increased flinching in no wheel/no PGE treatment group compared to vehicle treated or PGE2/wheel treated mice after 28 days of wheel access. These results indicate an analgesic effect of wheel exercise on PGE2 induced chronic von Frey hyperalgesia and capsaicin induced flinching, but not on daily distance running. 

Another experiment started with a 14 day course of daily PGE2 or vehicle intraplantar injections, then 14 days of running wheel access or no wheel access. There was no difference in the distance run/day between PGE2 and vehicle. Von Frey hindpaw threshold testing was performed weekly during the entire time period and for 14 days after the wheel access ended. The PGE2 mice developed von Frey hyperalgesia by day 7 of injections and wheel exercise starting at the end of the PGE2 treatment partially reversed this hyperalgesia and this effect persisted for 14 days after stopping wheel exercise. Capsaicin intraplantar injection caused no increased flinching in PGE2/wheel treated mice 14 days after wheel access. 

Another experiment started with a 14 day course of daily PGE2 or vehicle intraplantar injections, then 14 days of no treatment, then 14 days of running wheel access or no wheel access. There was no difference in the distance run/day between PGE2 and vehicle. The PGE2 mice developed von Frey hyperalgesia by day 7 of injections and wheel exercise starting at 14 days after the end of the PGE2 treatment partially reversed this hyperalgesia. Capsaicin intraplantar injection caused no increased flinching PGE2/wheel treated mice after 14 days of wheel access. 

Another experiment provided the mice with 28 days wheel access or no wheel access, starting at day 1, then after 8 days of wheel or no wheel access PGE2 or vehicle daily intraplantar injections were started for 14 days. The PGE2 mice did display a reduction in daily wheel running distance relative to the vehicle group. The PGE2/no wheel mice developed von Frey hyperalgesia by day 7 of PGE2 injections,but the PGE2/ wheel exercise group did not develop hyperalgesia. Capsaicin intraplantar injection caused no increased flinching PGE2/wheel treated mice after 14 days of wheel access. 

Theses results are novel and the experiments seem to be well performed but there are several issues. 

Major Concerns:

1. Repeated testing over time within the same groups of mice need to be analyzed with repeated measures 2 way ANOVA with Sidak post hoc testing.

The authors agree with the reviewer and, in fact, we already had analyzed repeated tests using 2-way ANOVA (Bonferroni as post hoc test), although this information was not clear in the original manuscript. Considering this comment, in the revised manuscript we clarified the test used (2-way ANOVA) and applied Sidak as post hoc test, removing multiple T-tests analysis.

2 The introduction and discussion need extensive revision with an English speaking editor.

The manuscript has been revised to correct grammatical mistakes.

3. There are many paragraphs in the introduction and discussion that are tangential or do not add to hypothesis or data, would recommend deleting lines 45-59, 76-78, 384-393, 418-422, 459-470.

As suggested by the reviewer, in the revised manuscript we deleted and/or altered some of these sentences.

4. There is no discussion regarding the mechanisms by which repeated PGE2 injections cause long lasting hyperalgesia. Dr Parada coauthored many papers on this topic with Dr Ferreira and it is surprising there is no discussion on how PGE2 injections cause long acting hyperalgesia by up-regulating the Nav1.8 sodium channels in the DRG, that PGE2 hyperalgesia is mediated by nonpeptidergic small sensory neurons that normally do not contribute to von Frey thresholds, and most importantly, that PGE2 induced mechanical hyperalgesia is mediated by DRG glutamate release acting on the NMDA receptor on DRG satellite cells. DRG satellite cells are immune cells capable of releasing proinflammatory cytokines and growth factors that may mediate chronic PGE2 sensitization.

We thank the reviewer for highlighting these aspects of the PGE2 model. Initially, the authors considered that including a discussion about the mechanisms associated with the model would make the session too much speculative and tangential, once we did not analyze molecular mechanism associated with the exercise-induced analgesia. However, considering this comment, we included in the revised manuscript a brief discussion speculating the possible effect of exercise on these known mechanisms (Discussion section; lines 406 to 416).

5. The discussion of the prior literature on exercise analgesic mechanisms in rodent pain models is superficial and incomplete. More discussion of exercise suppression of pronociceptive immune responses is required and this should be tied into the crucial role of DRG satellite immune cells in PGE2 chronic hyperalgesia. The following papers should be included and reviewed; Lopez-Alvarez et al, Pain 2015: 156: 1812-25; Almeida et al, Pain 2015:156:504- 513; Cobianchi et al, Neuroscience 2010: 168: 273-287; Shi et al, Anesthesiology 2018: 129: 557-75.

We rewrote some aspects of the discussion session, making efforts to include the suggested literature (Discussion section; lines 368 to 372, and 406 to 416). 

6. This is a descriptive paper with no mechanism or hypothesis of a mechanism for the analgesic effects of exercise.

Although our group believes that investigate molecular mechanisms associated with the antihyperalgesic effects of physical exercise is importante, in this study we did not aim to investigate such mechanisms. Thus, we did not present mechanisms associated with our behavioral data. This is undoubtedly a limitation of our study. However, based on the literature, we discuss several hypotheses of mechanisms for the analgesic effects of exercise. We discuss systemic aspects associated with the immune system and the production of cytokines and circulating opiods; we discuss neural aspects, both peripheral (DRG satellite cells) and central, and production of neurotransmitters and neurotrophic factors. In the revised manuscript the discussion on this issue was expanded and improved thanks to the reviwer’s comment #5. Also, the wide variety of mechanisms discussed in the manuscript illustrates well how exercise is a behavior that engages the entire body and its effects are systemic and multiple, involving several tissues and countless molecular agents that act synergistically. Therefore, the lack of mechanistic data in the present study is minimized. Furthermore, aspects such as the chronic pain model used in the study, which mimicks the main types of pain in the clinical setting; the bypass of placebo effect, that is a confounding factor on clinical trials of non-pharmacological interventions studies; the long-lasting antihyperalgesic effect of physical exercise, as well as, its expression even with the late onset of the intervention; and the scarce reported preventive effect, render our study robust and relevant, dispite some limitations. 

Minor concerns:

1. Need to make clear in methods if mice were housed in individual cages and if they had the wheel in their cage 24 hours/day. 

In the revised manuscript we included a better description of the methodology. As requested by the reviewer, we included the information that, after starting the experimental protocol, all mice remained single housed with one wheel each available 24h/day.

2. There is no information on the running wheels, vendor, or computerized data collection methods for measuring distance run/day?

In fact, this information was missing, and we added in the revised manuscript (“Voluntary running wheel exercise” section, Methods). 

3. Why did the authors inject the same amount and volume of PGE2 in mice as they previously used in rats, shouldn’t smaller doses and volumes be used in much smaller animals, please add to the discussion in a section on the limitations of the study.

As described by Ferreira et al (1990) the dose and volume of PGE2 injection for rats is 100ng/100ul/paw and, as described by Villarreal (2009), the volume for mice is 90ng/25ul/paw. In the present study we used the protocol described for mice by Villarreal and colleagues (2009) and, in the revised manuscript, we corrected the dose and volume informations (“Persistent hyperalgesia induction model” section, Methods). 

4. All figure legends need to be on one page.

In the revised manuscript we are showing all figure legends on one page.

---

## [Decision Letter · Decision Letter 1]

12 Aug 2020

PONE-D-20-03419R1

Running wheel exercise induces therapeutic and preventive effects on inflammatory stimulus-induced persistent hyperalgesia in mice.

PLOS ONE

Dear Dr. Sartori,

Thank you for submitting your manuscript to PLOS ONE. After careful consideration, we feel that it has merit but does not fully meet PLOS ONE’s publication criteria as it currently stands. Therefore, we invite you to submit a revised version of the manuscript that addresses the points raised during the review process.

Only one of the original reviewers were available to review your revision (see below for their comments). Both the reviewer and I agree that the manuscript reports a series of interesting studies. However, the Discussion lacks some critical view and insights into possible mechanisms. Currently the Discussion is mostly repetition of the results and how the results confirm some of the past literature. This does not meet the scientific standards required in PLOS ONE. Therefore, I would like to encourage authors to discuss possible mechanisms underlying their findings. Additionally, as the authors acknowledge that the  molecular mechanisms is important, please include this in your discussion.

We look forward to receiving your revised manuscript.

Kind regards,

Amir-Homayoun Javadi, PhD

Academic Editor

PLOS ONE

Reviewers' comments:

Reviewer's Responses to Questions

**Comments to the Author**

1. If the authors have adequately addressed your comments raised in a previous round of review and you feel that this manuscript is now acceptable for publication, you may indicate that here to bypass the “Comments to the Author” section, enter your conflict of interest statement in the “Confidential to Editor” section, and submit your "Accept" recommendation.

Reviewer #1: All comments have been addressed

2. Is the manuscript technically sound, and do the data support the conclusions?

Reviewer #1: No

3. Has the statistical analysis been performed appropriately and rigorously? 

Reviewer #1: Yes

4. Have the authors made all data underlying the findings in their manuscript fully available?

Reviewer #1: No

5. Is the manuscript presented in an intelligible fashion and written in standard English?

Reviewer #1: Yes

6. Review Comments to the Author

Reviewer #1: (No Response)

7. PLOS authors have the option to publish the peer review history of their article (what does this mean?). If published, this will include your full peer review and any attached files.

Reviewer #1: No

---

## [Author Response · Author response to Decision Letter 1]

3 Sep 2020

Authors: First, we would like to thank the contributions made by the editor and reviewer #1. Below are our comments, point-by-point, to the comments and questions raised. We are at your disposal for any additional clarification that may be required.

#Editor: “Dear Dr. Sartori,

Thank you for submitting your manuscript to PLOS ONE. After careful consideration, we feel that it has merit but does not fully meet PLOS ONE’s publication criteria as it currently stands. Therefore, we invite you to submit a revised version of the manuscript that addresses the points raised during the review process.

Only one of the original reviewers were available to review your revision (see below for their comments). Both the reviewer and I agree that the manuscript reports a series of interesting studies. However, the Discussion lacks some critical view and insights into possible mechanisms. Currently the Discussion is mostly repetition of the results and how the results confirm some of the pastliterature. This does not meet the scientific standards required in PLOS ONE. Therefore, I would like to encourage authors to discuss possible mechanisms underlying their findings.”

Authors: We appreciate the comment and, in the revised manuscript, we added some critical view and insights into possible mechanism behind our findings. We better discussed the anti-inflammatory effect of physical exercise and its effects on endogenous PGE2. Also, we discussed the possible involvement of the serotonergic and opioidergic system in the exercise-induced analgesia. We also improved the discussion regarding the effects of exercise on DRG. Finally, we highlighted the effect of exercise on oxidative stress and the endocannabinoid system as possible mechanisms.

“Additionally, as the authors acknowledge that the molecular mechanisms is important, please include this in your discussion.”

Authors: We included this statement in the revised manuscript.

“Please submit your revised manuscript by Sep 26 2020 11:59PM. If you will need more time than this to complete your revisions, please reply to this message or contact the journal office at plosone@plos.org. Please include the following items when submitting your revised manuscript:

• A rebuttal letter that responds to each point raised by the academic editor and reviewer(s). You should upload this letter as a separate file labeled 'Response toReviewers'.

We look forward to receiving your revised manuscript.

Kind regards,

Amir-Homayoun Javadi, PhD

Academic Editor

PLOS ONE”

Authors: Appreciated.

“Reviewers' comments:

Reviewer's Responses to Questions

Comments to the Author

1. If the authors have adequately addressed your comments raised in a previous round of review and you feel that this manuscript is now acceptable for publication, you may indicate that here to bypass the “Comments to the Author” section, enter your conflict of interest statement in the “Confidential to Editor” section, and submit your "Accept" recommendation.

Reviewer#1: All comments have been addressed”

Authors: Appreciated

“2. Is the manuscript technically sound, and do the data support the conclusions?

Reviewer #1: No”

Authors: In the revised manuscript we made few changes in our conclusion to directly refer to our findings. 

“3. Has the statistical analysis been performed appropriately and rigorously?

Reviewer #1: Yes”

Authors: Appreciated

“4. Have the authors made all data underlying the findings in their manuscript fully available?

The PLOS Data policy requires authors to make all data underlying the findings described in their manuscript fully available withoutrestriction, with rare exception (please refer to the Data Availability Statement in the manuscript PDF file). The data should be provided as part of the manuscript or its supporting information, or deposited to a public repository. For example, in addition to summary statistics, the data points behind means, medians and variance measures should be available. If there are restrictions on publicly sharing data—e.g. participant privacy or use of data from a third party—those must be specified.

Reviewer #1: No”

Authors: There is a statement declaring that “All data presented in the current manuscript can be obtained from the corresponding author”. 

“5. Is the manuscript presented in an intelligible fashion and written in standard English?

Reviewer #1: Yes”

Authors: Appreciated

“6.Review Comments to the Author

Reviewer #1: (No Response)”

Authors: Appreciated

“7. PLOS authors have the option to publish the peer review history of their article (what does this mean?). If published, this will include your full peer review and any attached files.

Do you want your identity to be public for this peer review? For information about this choice, including consentwithdrawal, please see our Privacy Policy.

Reviewer #1: No”

Authors: Appreciated

---

## [Editor Report · Decision Letter 2]

21 Sep 2020

Running wheel exercise induces therapeutic and preventive effects on inflammatory stimulus-induced persistent hyperalgesia in mice.

PONE-D-20-03419R2

Dear Dr. Sartori,

We’re pleased to inform you that your manuscript has been judged scientifically suitable for publication and will be formally accepted for publication once it meets all outstanding technical requirements.

Kind regards,

Amir-Homayoun Javadi, PhD

Academic Editor

PLOS ONE

---

## [Editor Report · Acceptance letter]

28 Sep 2020

PONE-D-20-03419R2 

Running wheel exercise induces therapeutic and preventive effects on inflammatory stimulus-induced persistent hyperalgesia in mice. 

Dear Dr. Sartori:

I'm pleased to inform you that your manuscript has been deemed suitable for publication in PLOS ONE. Congratulations! Your manuscript is now with our production department. 

Kind regards, 

on behalf of

Dr. Amir-Homayoun Javadi 

Academic Editor

PLOS ONE